# Clinical Utility of Personalized Serum IgG Subclass Ratios for the Differentiation of IgG4-Related Sclerosing Cholangitis (IgG4-SC) from Primary Sclerosing Cholangitis (PSC) and Cholangiocarcinoma (CCA)

**DOI:** 10.3390/jpm12060855

**Published:** 2022-05-24

**Authors:** Jae Keun Park, Dongwuk Kim, Jeong Min Lee, Kwang Hyuck Lee, Kyu Taek Lee, Joo Kyung Park, Jong Kyun Lee

**Affiliations:** 1Department of Internal Medicine, Kangnam Sacred Heart Hospital, Hallym University College of Medicine, Seoul 07441, Korea; hanyangjj@gmail.com; 2Division of Gastroenterology, Department of Medicine, Samsung Medical Center, Sungkyunkwan University School of Medicine, Seoul 06351, Korea; wjdtlr57@naver.com (D.K.); mdmiya@naver.com (J.M.L.); lkhyuck@gmail.com (K.H.L.); ktcool.lee@samsung.com (K.T.L.); jksophie.park@samsung.com (J.K.P.); 3Department of Clinical Research Design and Evaluation, Samsung Advanced Institute for Health Sciences & Technology (SAIHST), Sungkyunkwan University, Seoul 16419, Korea

**Keywords:** IgG4-related sclerosing cholangitis, IgG4, IgG subclass, primary sclerosing cholangitis, cholangiocarcinoma

## Abstract

Background: The differential diagnosis of immunoglobulin G4-sclerosing cholangitis (IgG4-SC) from primary sclerosing cholangitis (PSC) or cholangiocarcinoma (CCA) is important. In this study, we aimed to find the best combinations of serum IgG subclasses and IgG4 levels for differentiating IgG4-SC from PSC or CCA. Methods: In total, 31 patients with IgG4-SC, 27 patients with PSC, and 40 patients with CCA were enrolled from 2003 to 2017 at a single tertiary referral center. We retrospectively assessed the IgG4, IgG4/IgG1, IgG4/(IgG1+IgG3), and (IgG4+IgG2)/(IgG1+IgG3) in each of the patients. ROC curves were established to obtain the optimal cutoff value for each parameter. McNemar’s test was used to compare the sensitivities, specificities, and accuracies of diagnostic algorithms. Results: In differentiating IgG4-SC from PSC, the accuracies of IgG4/IgG1 ≥ 0.087 and of IgG4/(IgG1+IgG3) ≥ 0.081 were significantly higher than that of IgG4 ≥ 135 mg/dL alone (78% vs. 66%, *p* = 0.025). Serum IgG4 ≥ 52 mg/dL showed the best accuracy for differentiation of IgG4-SC from CCA, with a sensitivity and specificity of 80% and 82%, respectively, but this was statistically not significant (*p* = 0.405). Conclusions: The serum IgG4/IgG1 or IgG4/(IgG1+IgG3) level may help to differentiate IgG4-SC from PSC. IgG4 alone is the most accurate serologic marker for the differentiation of IgG4-SC from CCA.

## 1. Introduction

IgG4-related sclerosing cholangitis (IgG4-SC) is a biliary manifestation of IgG4-related systemic disease [1,2]. IgG4-SC is often characterized by the biliary stricture, and this can be mimicked by other biliary diseases such as primary sclerosing cholangitis (PSC) or cholangiocarcinoma (CCA) [3]. Thus, the differential diagnosis of IgG4-SC from PSC or CCA is important, because biliary strictures in IgG4-SC usually respond to steroid therapy [2]. There have been some diagnostic criteria proposed for IgG4-SC [4]. These diagnostic criteria are composed of serologic, radiologic, and histologic features. All proposed criteria suggest elevated serum IgG4 levels as part of the diagnostic criteria [5,6,7]. As a result, an elevated serum IgG4 level, especially serum IgG4 ≥ 135 mg/dL, is generally thought of as an important criterion for diagnosing IgG4-SC. However, serum IgG4 sometimes increases in PSC. Mendes et al. reported that 9% of their PSC patients showed serum IgG4 > 140 mg/dL [8], while Oseini et al. reported that 22.6% of the 31 PSC patients among their 126 CCA patients had elevated serum IgG4 levels [9].

Serum IgG, the predominant isotype in the human body, is divided into four subclasses: IgG1, IgG2, IgG3, and IgG4 [10]. Among these four subclasses, IgG1 is the most abundant of the total IgG in normal subjects [11]. The ratio for each IgG subclass to total IgG is 65% (IgG1), 25% (IgG2), 6% (IgG3), and 4% (IgG4). In general, the IgG subclass/total IgG ratio and quantity of IgG subclasses remain constant. However, the IgG subclass/total IgG ratio and quantity of IgG subclasses typically change in some diseases. For example, in IgG4-related diseases, several studies have revealed that the total IgG, IgG1, IgG2, IgG4, and IgE ratios are usually increased [12,13,14,15]. A recent article by Boonstra et al. suggested that the IgG subclass could be used to distinguish IgG4-SC from PSC [16]. This report showed that in patients with moderately elevated serum IgG4 (140 < IgG4 < 280 mg/dL), an IgG4/IgG1 ratio cutoff value of 0.24 could help in the differentiation of IgG4-SC from PSC. Additionally, Zhang et al. showed that some autoimmune diseases such as systemic sclerosis and Sjogren syndrome are associated with higher IgG1 and IgG3 and lower IgG2 levels than healthy controls [17]. These findings suggest that other IgG subclass combinations rather than IgG4 alone could help distinguish IgG4-SC from PSC or CCA. Thus, in this study, we aimed to find other better combinations of serum IgG subclasses and IgG4 levels to differentiate IgG4-SC from PSC or CCA.

## 2. Materials and Methods

### 2.1. Study Population

We analyzed 31 patients with IgG4-SC, 27 patients with PSC, and 40 patients with CCA who underwent IgG subclass from 2003 to 2017 at Samsung Medical Center. IgG4-SC patients with autoimmune pancreatitis (AIP) were excluded from the analysis. Moreover, biopsy-proven CCA patients who had distant metastasis, biliary strictures, and vascular invasion at the time of diagnosis were excluded from the analysis. This study was conducted according to the Declaration of Helsinki and approved by the Institutional Review Board of Samsung Medical Center, Seoul, Korea (No. SMC 2017-09-122).

### 2.2. Data Collection and Definitions

We collected patient demographic data and IgG subclass data. We also reviewed medical charts, pathology results, clinical outcomes, and image findings. IgG4-SC was diagnosed based on the criteria proposed by the Research Committee of IgG4-Related Diseases and the Research Committee of Intractable Diseases of Liver and Biliary Tract, in association with the Ministry of Health, Labor, and Welfare, Japan, and the Japan Biliary Association in 2012 [6].

PSC was diagnosed according to several characteristics, such as (1) clinical symptoms (fever, pruritus, and right upper abdominal quadrant pain); (2) cholestatic laboratory profile, such as elevated alkaline phosphatase (ALP), gamma-glutamyl transferase (GGT), and bilirubin level; (3) bile duct image findings with multifocal strictures and segmental dilatations using endoscopic retrograde cholangiopancreatography (ERCP) or magnetic resonance cholangiopancreatography (MRCP); and (4) pathologic profile (in some patients) [16]. Secondary sclerosing cholangitis was excluded in all patients. Cholangiocarcinoma was diagnosed with combinations of clinical features, radiologic findings, pathologic profile by surgery, and/or biomarkers.

### 2.3. Statistical Analysis

We used the Kruskal–Wallis test to compare categorical and continuous data between groups. Receiver operator characteristic (ROC) curves were used to confirm optimal cutoff values of IgG subclass combinations and serum IgG4 to differentiate IgG4-SC from PSC or CCA. McNemar’s test was used to compare the sensitivities, specificities, and accuracies of the diagnostic algorithms. Additionally, Bennett’s test was used to compare the positive predictive value (PPV) and the negative predictive value (NPV). A *p*-value of <0.05 was considered statistically significant. Statistical analyses were performed using SAS version 9.4 (SAS Institute, Cary, NC, USA) and SPSS Statistics 23 (IBM, Chicago, IL, USA).

## 3. Results

### 3.1. Clinical Characteristics of the Study Population

In Table 1, the clinical characteristics of the study population are listed. A total of 31 IgG4-SC, 27 PSC, and 40 CCA patients who checked an IgG subclass at diagnosis were included in this study. The mean age of diagnosis was significantly different among the three groups. The mean age was 65.0 ± 11.9 years in the IgG4-SC group, 50.3 ± 13.4 years in the PSC group, and 63.5 ± 8.8 years in the CCA group (*p* = 0.008). Sex did not show a statistically significant difference. Except for IgG3, all IgG subclasses showed statistically significant differences among the three groups. The mean IgG1 (mg/dL) level was 962.1 ± 460.4 in the IgG4-SC group, 846.7 ± 327.0 in the PSC group, and 654.9 ± 239.8 in the CCA group (*p* = 0.001). The mean IgG2 (mg/dL) level was 743.8 ± 343.9 in the IgG4-SC group, 586.2 ± 252.3 in the PSC group, and 511.0 ± 198.2 in the CCA group (*p* = 0.007). The mean IgG3 (mg/dl) level was 80.5 ± 79.8 in the IgG4-SC group, 74.2 ± 47.8 in the PSC group, and 42.5 ± 25.8 in the CCA group (*p* = 0.053). The mean IgG4 (mg/d:) level was 226.5 ± 184.2 in the IgG4-SC group, 37.3 ± 20.1 in the PSC group, and 46.5 ± 48.7 in the CCA group (*p* < 0.001). Elevated IgG4 levels (>135 mg/dL) were noted in 17 patients (54.8%) in the IgG4-SC group and two patients (5%) in the CCA group. The PSC group patients did not show elevated IgG4 levels (*p* < 0.001).

### 3.2. Performance of the Serum IgG Subclass Combinations for the Differential Diagnosis of IgG4-SC from PSC

The comparison of the diagnostic performance of each IgG subclass combination between IgG4-SC and PSC is demonstrated in Table 2. A serum IgG4 level of ≥135 mg/dL yielded a sensitivity of 54% (95% CI: 37–70) with a specificity of 100% (95% CI: 74–100) for IgG4-SC. The PPV was 100% (95% CI: 81–100), whereas the NPV was 44% (95% CI: 26–62). In this study, we analyzed the optimal cutoff value of serum IgG4 (68 mg/dL). An IgG4 level of ≥68 mg/dL yielded a sensitivity of 64% (95% CI: 46–78) with a specificity of 100% (95% CI: 74–100) for IgG4-SC. PPV was 100% (95% CI: 83–100), whereas NPV was 50% (95% CI: 30–69).

We compared IgG4/subclass ratios to distinguish IgG4-SC from PSC by using ROC curves. The IgG4/subclass ratios consisted of the following groups: IgG4/IgG1, IgG4/(IgG1+IgG3), and (IgG4+IgG2)/(IgG1+IgG3). The largest areas under the curve (AUC) reached the optimal combination of sensitivity and specificity at 0.087, 0.081, and 1.159 in IgG4/IgG1, IgG4/(IgG1+IgG3), and (IgG4+IgG2)/(IgG1+IgG3), respectively. In terms of accuracy, the IgG4/IgG1 and IgG4/(IgG1+IgG3) groups were found to be the highest in terms of sensitivity (70%) and specificity (100%), with statistically significantly higher sensitivity and accuracy than IgG4 ≥ 135 mg/dL (*p* = 0.025).

### 3.3. Performance of the Serum IgG Subclass Combinations for the Differential Diagnosis of IgG4-SC from CCA

The comparison of the diagnostic performance of each IgG subclass combination between IgG4-SC and CCA is demonstrated in Table 3. A serum IgG4 level of ≥135 mg/dL yielded a sensitivity of 54% (95% CI: 37–70) with a specificity of 95% (95% CI: 83–98) for IgG4-SC. The PPV was 89% (95% CI: 68–97), whereas the NPV was 73% (95% CI: 59–83). We analyzed the ROC curve to confirm the optimal cutoff value of serum IgG4 (52 mg/dL). IgG4 ≥ 52 mg/dL yielded a sensitivity of 80% (95% CI: 63–90) with a specificity of 82% (95% CI: 68–91) for IgG4-SC. The PPV was 78% (95% CI: 61–88), whereas the NPV was 84% (95% CI: 70-–92). Comparing the IgG4/subclass ratios to distinguish IgG4-SC from CCA using the ROC curves, the largest AUC reached the optimal combination of sensitivity and specificity at 0.087, 0.081, and 1.159 in IgG4/IgG1, IgG4/(IgG1+IgG3), and (IgG4+IgG2)/(IgG1+IgG3), respectively. In terms of accuracy, IgG4 ≥ 52 mg/dL was found to be the highest with the highest sensitivity (80%) and a statistically significantly higher sensitivity than IgG4 ≥ 135 mg/dL (*p* = 0.005). However, the accuracy of IgG4 ≥ 52 mg/dL was not statistically significant compared to IgG4 ≥ 135 mg/dL (*p* = 0.405). Plus, the specificity of IgG4 ≥ 52 mg/dL was much lower than that of IgG4 ≥ 135 mg/dL (IgG4 ≥ 52 mg/dL vs. IgG4 ≥ 135 mg/dL; 82% (95% CI: 68–91) vs. 95% (95% CI: 83–98); *p* = 0.025).

### 3.4. ROC Curves of the IgG Subclass Combinations for the Diagnosis of IgG4-SC from PSC

Table 4 and Figure 1A show the AUCs (95% CIs) of the serum IgG subclass combinations for differentiation of IgG4-SC from PSC. IgG4 ≥ 68 mg/dL, IgG4/IgG1 ≥ 0.087, and IgG4/(IgG1+IgG3) ≥ 0.081 were almost equally predictive of IgG4-SC. (IgG4+IgG2)/(Ig G1+IgG3) ≥ 1.159 had the lowest AUC for IgG4-SC (AUC: 0.730; 95% CI:0.572–0.888), while IgG4/(IgG1+IgG3) ≥ 0.081 showed the highest AUC value for IgG4-SC (AUC: 0.853; 95% CI:0.739–0.967). The AUCs of IgG4 ≥ 68 mg/dL and IgG4/IgG1 ≥ 0.087 were 0.827 (95% CI: 0.704–0.950) and 0.848 (95% CI: 0.732–0.963), respectively.

### 3.5. ROC Curves of the IgG Subclass Combinations for the Diagnosis of IgG4-SC from CCA

The AUCs (95% CIs) of the serum IgG subclass combinations for the differentiation of IgG4-SC from PSC are presented in Table 5 and Figure 1B. In our study, IgG4 ≥ 52 mg/dL had the highest AUC values for IgG4-SC (AUC: 0.826; 95% CI: 0.717–0.935), while (IgG4+IgG2)/(Ig G1+IgG3) ≥ 1.159 had the lowest AUCs for IgG4-SC (AUC: 0.617; 95% CI: 0.478–0.757). IgG4/IgG1 ≥ 0.087 and IgG4/(IgG1+IgG 3) ≥ 0.081 were almost equally predictive of IgG4-SC. The AUCs of IgG4/IgG1 ≥ 0.087 and IgG4/(IgG1+IgG 3) ≥ 0.081 were 0.760 (95% CI: 0.639–0.881) and 0.753 (95% CI: 0.629–0.876), respectively.

## 4. Discussion

In this study, we assessed the best combination of serum IgG subclasses and IgG4 levels for differentiating IgG4-SC from PSC or CCA. According to the AUCs of the serum IgG subclass combinations, IgG4/(IgG1+IgG3) ≥ 0.081 was found to be the most relevant combination for distinguishing IgG4-SC from PSC. The accuracy of IgG4 ≥ 135 mg/dL (66%; 95% CI: 51–78) was lower than that of IgG4/(IgG1+IgG3) ≥ 0.081 (78%; 95% CI: 64–88). On the contrary, IgG4 ≥ 52 mg/dL was the most relevant combination of serum IgG subclasses and IgG4 levels for distinguishing IgG4-SC from CCA. Even though the accuracy of IgG4 ≥ 52 mg/dL (81%; 95% CI: 71–88) was higher than that of IgG4 ≥ 135 mg/dL (77%; 95% CI: 66–85) for differentiating IgG4-SC from CCA, the specificity of IgG4 ≥ 52 mg/dL (82%; 95% CI: 68–91) was lower than that of IgG4 ≥ 135 mg/dL (95%; 95% CI: 83–98). In clinical practice, it is more important to exclude patients with CCA from other benign diseases. Therefore, the diagnostic value of IgG4 ≥ 52 mg/dL seems less practical than the preexisting IgG4 ≥ 135 mg/dL as a serologic marker.

Three representative diagnostic criteria for IgG4-SC have been proposed [5,6,7]. It is important to note that these diagnostic criteria commonly refer to an increased level of serum IgG4. In fact, several previous studies have found that a significant elevation of the serum IgG4 level in IgG4-SC is a useful serologic marker for distinguishing AIP from other pancreatic diseases [2,6,8,12,18]. In AIP, a 135 mg/dL serum IgG4 cutoff level has been commonly used as a diagnostic criterion [4].

In IgG4-SC, only a small number of studies have reported the cutoff level of serum IgG4 for distinguishing IgG4-SC from other biliary diseases, such as PSC and CCA, although a cutoff level of serum IgG4 ≥ 135 mg/dL has also been widely used. In a previous comparative study, Nakazawa et al. reported that the serum IgG4 based on cholangiography classification is a useful diagnostic criterion for distinguishing IgG4-SC from PSC and CCA [4]. However, the cutoff levels for serum IgG4 for distinguishing the above diseases have not been clearly defined.

Several previous studies have found a significant association between the elevated level of the serum IgG subclass (IgG2 and IgG4) and IgG4-related disease (IgG4-RD) [14,16,19,20]. In patients with PSC, one previous study reported that 23% of the study patients had hypergammaglobulinemia [21]. Other studies have reported that elevated levels of IgG4 in PSC are associated with cirrhosis and poor disease course [8,22]. Several cases among the enrolled patients of these studies showed clinical features related to IgG4-RD. On the contrary, in the cases of CCA patients, no data are available concerning the association between the CCA and IgG subclasses except IgG4. One report revealed that an elevated IgG4 subclass in CCA is associated with PSC [9].

In our study, patients with IgG4-SC presented elevated mean values in the IgG2 and IgG4 subclasses. In contrast, the mean values of the serum IgG subclasses were not elevated in those patients with PSC and CCA. These differences may have resulted for several possible immunopathological and genetic reasons. First, similarity of the molecular mass (146 kD), number of amino acids in the hinge region (12 amino acids), and affinity to the Fc receptor between the IgG2 and IgG4 subclasses have been reported [22]. These similarities may have an impact on the elevated serum level of the IgG2 subclass. Second, the similarity of antibody responses has also been revealed in a previous study [10]. Unlike IgG1 and IgG3, polysaccharide antigens are associated with IgG2 and IgG4. In particular, long-term repeated antigen exposure often causes the formation of IgG4 antibodies [23]. In addition, a previous study documented a low serum IgG2 level often found in patients with IgG4 deficiency [24]. Third, the genetic linkage related to the IgG2 and IgG4 coding gene may provide a possible explanation. As a matter of fact, it was already known that the IgG2 and IgG4 coding gene (gene fragments Cγ2 and Cγ4, respectively) reside side by side [14].

In a recent study, Boonstra et al. documented that only 28% of their study patients with IgG4-SC showed elevated serum IgG4 levels ranging between 140 and 280 mg/dL [16]. Several previous studies have reported that serum IgG4 levels are also elevated in 9–27% of patients with PSC [8,22,25,26,27,28]. Furthermore, Oseini et al. assessed the utility of the serum IgG4 level in distinguishing IgG4-SC from CCA [9]. The authors reported that 13.5% of their 126 CCA patients showed a serum IgG4 level >140 mg/dL, while 3.2% revealed a serum IgG4 level >280 mg/dL. In addition, measuring the serum IgG4 levels in nine Japanese high-volume centers revealed that 8.1% of CCA patients had higher IgG4 levels than the cutoff level [28]. Therefore, distinguishing IgG4-SC from PSC and CCA with the serum IgG4 level only was insufficient.

There are some limitations to our present study. First, our study data were collected retrospectively, resulting in the potential for selection bias. Thus, prospective validation cohort studies are required to confirm our results. The second limitation is the small number of study patients, especially patients with PCS (*n* = 27). The reason for this small number of patients with PCS is because PSC is rare in East Asia [29,30]. Therefore, to overcome the above limitation, a nationwide multi-center study is necessary. Third, CCA can occur in patients with IgG4-SC due to chronic inflammation of the biliary duct and cholestasis, which are risk factors of CCA. However, only a few case reports demonstrated an association between IgG4-SC and CCA [31,32]. The longitudinal relationship between IgG4-SC and CCA should be examined in future studies. Fourth, this study focuses on the combinations of serum IgG subclasses and IgG4 levels due to missing values in the clinical and biochemical data, but this is inherent to the retrospective study. Well-designed randomized prospective studies are needed to overcome this limitation of this study.

IgG4/(IgG1+IgG3) ≥ 0.081 showed the strongest predictive power for distinguishing IgG4-SC from PSC. In distinguishing IgG4-SC from CCA, IgG4 ≥ 52 mg/dL showed the strongest predictive power. Our findings indicate that the novel measurement of serum IgG4/IgG1 or IgG4/(IgG1+IgG3) level may help to differentiate IgG4-SC from PSC. Additionally, IgG4 alone was shown to be the most accurate serologic marker for the differentiation of IgG4-SC from CCA. However, further large-scale multicenter prospective studies are required to confirm the suitability of the use of these combinations of serum IgG subclasses and IgG4 levels, as well as our results.

## Figures and Tables

**Figure 1 jpm-12-00855-f001:**
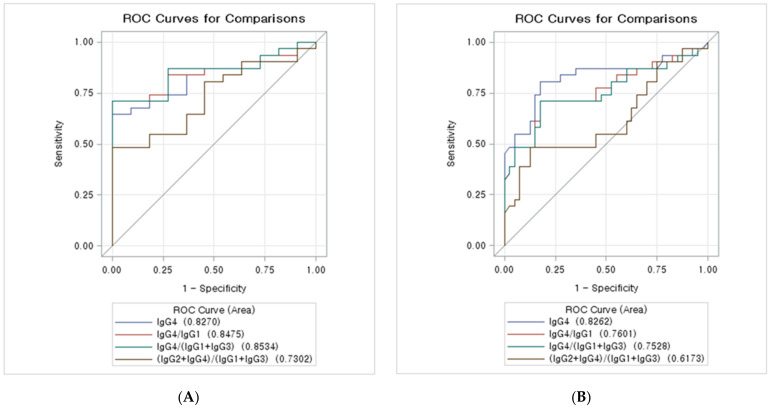
ROC curves of the IgG subclass combinations: (**A**) ROC curves of the IgG subclass combinations for diagnosis of IgG4-SC from PSC. The ROC curve of IgG4 ≥ 68 mg/dL, IgG4/IgG1 ≥ 0.087, IgG4/(IgG1+IgG3) ≥ 0.081, and (IgG4+IgG2)/(IgG1+IgG3) ≥ 1.159 for comparison. (**B**) ROC curves of the IgG subclass combinations for the diagnosis of IgG4-SC from CCA. The ROC curve of IgG4 ≥ 52 mg/dL, IgG4/IgG1 ≥ 0.087, IgG4/(IgG1+IgG3) ≥ 0.081, and (IgG4+IgG2)/(IgG1+IgG3) ≥ 1.159 for comparison. IgG, immunoglobulin G.

**Table 1 jpm-12-00855-t001:** Baseline characteristics.

Variables	IgG4-SC	PSC	CCA	*p*-Value
Number of groups	31	27	40	
Age (years)	65.0 ± 11.9	50.3 ± 13.4	63.5 ± 8.89	0.008
Male	25 (80.6)	14 (51.9)	23 (57.5)	0.090
IgG1 (mg/dL)	962.1 ± 460.4	846.7 ± 327.0	654.9 ± 239.8	0.001
IgG2 (mg/dL)	743.8 ± 343.9	586.2 ± 252.3	511.0 ± 198.2	0.007
IgG3 (mg/dL)	80.5 ± 79.8	74.2 ± 47.8	42.5 ± 25.8	0.053
IgG4 (mg/dL)	226.5 ± 184.2	37.3 ± 20.1	46.5 ± 48.7	<0.001
IgG4 ≥ 135 mg/dL	17 (54.8)	0 (0)	2 (5)	<0.001

IgG4-SC, IgG4-related sclerosing cholangitis; CCA, cholangiocarcinoma; IgG, immunoglobulin G. Data are shown as the mean ± s.d. or number (%) of patients.

**Table 2 jpm-12-00855-t002:** Performance of the serum IgG subclass combinations for the differentiation of IgG4-SC from PSC.

Variables (%)	IgG4 ≥ 135 mg/dL	IgG4 ≥ 68 mg/dL	IgG4/IgG1 ≥ 0.087	IgG4/(IgG1+IgG3) ≥ 0.081	(IgG4+IgG2)/ (IgG1+IgG3) ≥ 1.159
Sensitivity (95% CI)	54 (37–70)	64 (46–78)	70 (53–83)	70 (53–83)	48 (31–65)
Specificity (95% CI)	100 (74–100)	100 (74–100)	100 (74–100)	100 (74–100)	100 (74–100)
PPV (95% CI)	100 (81–100)	100 (83–100)	100 (85–100)	100 (85–100)	100(79–100)
NPV (95% CI)	44 (26–62)	50 (30–69)	55 (34–74)	55 (34–74)	40 (24–59)
Accuracy (95% CI)	66 (51–78)	73 (58–84)	78 (64–88)	78 (64–88)	61 (46–75)

IgG4-SC, IgG4-related sclerosing cholangitis; PSC, primary sclerosing cholangitis; IgG, immunoglobulin G; CI, confidence interval; PPV, positive predicted value; NPV, negative predicted value.

**Table 3 jpm-12-00855-t003:** Performance of the serum IgG subclass combinations for the differentiation of IgG4-SC from CCA.

Variables (%)	IgG4 ≥ 135 mg/dL	IgG4 ≥ 52 mg/dL	IgG4/IgG1 ≥ 0.087	IgG4/(IgG1+IgG3) ≥ 0.081	(IgG4+IgG2)/ (IgG1+IgG3) ≥ 1.159
Sensitivity (95% CI)	54 (37–70)	80 (63–90)	70 (53–83)	70 (53–83)	48 (31–65)
Specificity (95% CI)	95 (83–98)	82 (68–91)	82 (68–91)	82 (68–91)	87 (73–94)
PPV (95% CI)	89 (68–97)	78 (61–88)	75 (57–87)	75 (57–87)	75 (53–88)
NPV (95% CI)	73 (59–83)	84 (70–92)	78 (64–88)	78 (64–88)	68 (54–79)
Accuracy (95% CI)	77 (66–85)	81 (71–88)	77 (66–85)	77 (66–85)	70 (58–79)

IgG4-SC, IgG4-related sclerosing cholangitis; CCA, cholangiocarcinoma; IgG, immunoglobulin G; CI, confidence interval; PPV, positive predicted value; NPV, negative predicted value.

**Table 4 jpm-12-00855-t004:** Area under the curve of serum IgG subclass combination for the differentiation of IgG4-SC from PSC.

Variables	AUC (95% CI)
IgG4 ≥ 68 mg/dL	0.827 (0.704–0.950)
IgG4/IgG1 ≥ 0.087	0.848 (0.732–0.963)
IgG4/(IgG1+IgG3) ≥ 0.081	0.853 (0.739–0.967)
(IgG4+IgG2)/(IgG1+IgG3) ≥ 1.159	0.730 (0.572–0.888)

IgG4-SC, IgG4-related sclerosing cholangitis; CCA, cholangiocarcinoma; IgG, immunoglobulin G. Data are shown as the mean ± s.d. or number (%) of patients.

**Table 5 jpm-12-00855-t005:** Area under the curve of serum IgG subclass combinations for the differentiation of IgG4-SC from CCA.

Variables	AUC (95% CI)
IgG4 ≥ 52 (mg/dL)	0.826 (0.717–0.935)
IgG4/IgG1 ≥ 0.087	0.760 (0.639–0.881)
IgG4/(IgG1+IgG3) ≥ 0.081	0.753 (0.629–0.876)
(IgG4+IgG2)/(IgG1+IgG3) ≥ 1.159	0.617 (0.478–0.757)

IgG4-SC, IgG4-related sclerosing cholangitis; CCA, cholangiocarcinoma; IgG, immunoglobulin G. Data are shown as the mean ± s.d. or number (%) of patients.

## Data Availability

The raw data supporting the conclusion of this article will be made available by the authors, without undue reservation.

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
