# Peer review of "Clinical Utility of Personalized Serum IgG Subclass Ratios for the Differentiation of IgG4-Related Sclerosing Cholangitis (IgG4-SC) from Primary Sclerosing Cholangitis (PSC) and Cholangiocarcinoma (CCA)"

_jpm, 2022, doi:10.3390/jpm12060855_

Round 1
Reviewer 1 Report
Interesting topic, since not much is still known about exact IgG levels and ratios typical for these, still quite new, diseases. Therefore, data is more than necessary.
Author Response
COVER LETTER
09 May 2022
Editor-in-Chief
Journal of Personalized Medicine
Dear Editor-in-Chief
We would like to express our sincere thanks to you and the reviewers for the thorough review of our manuscript (Manuscript ID: jpm-1688832) titled “Clinical utility of personalized serum IgG subclass ratios for the differentiation of IgG4-related sclerosing cholangitis (IgG4-SC) from primary sclerosing cholangitis (PSC) and cholangiocarcinoma (CCA)” and for the opportunity to submit a revised and improved version. We believe that by addressing the concerns, we have considerably improved our manuscript. Below this letter, we have provided point-by-point responses to the reviewer’s comment.
We hope that you find the current version of the manuscript suitable for publication in your journal. We will certainly be willing to make additional changes should they be required.
Thank you for your consideration. I look forward to hearing from you.
Sincerely,
Jong Kyun Lee, MD, PhD
Departments of Medicine, Samsung Medical Center, Sungkyunkwan University School
of Medicine, 81 Irwon-ro, Gangnam-gu, Seoul, 06351, South Korea
Email: jongk.lee@samsung.com
+82-2-3410-3409
82-2-3410-6983
Response to Reviewer Comments
Reviewer 1
Interesting topic, since not much is still known about exact IgG levels and ratios typical for these, still quite new, diseases. Therefore, data is more than necessary.
Response: Thank you for your feedback. We will design further large-scale multicenter-based prospective studies that are also required to confirm the suitability of the use of these combinations of serum IgG subclass and IgG4 level and our results.

Reviewer 2 Report
The manuscript writtens by Jae Keun Park et al. investigates the best combinations of serum IgG subclass and IgG4 level for differentiating IgG4-SC from PSC or CCA. There is some interesting results where authors claimed that serum IgG4/IgG1 or IgG4/(IgG1+IgG3) level may help to differentiate IgG4-SC from PSC, and IgG4 alone is the most accurate serologic marker for the differentiation of IgG4-SC from CCA.
Regardless of the fact that it is an interesting work, some changes should be made in manuscript.
- In Statistical analysis the authors claimed that they used Kruskal-Willis test. This is not correct, it is Kruskal-Wallis test.
- Also, the main limitation of this study is the small number of the study patients. For serious conclusion it is necessary to include a significantly larger number of patients.
- It is necessary to correct the grammar of the English language.
- The results are very difficult to follow. Also, it is necessary to compare some additional clinical and biochemical data of patients for some serious conclusion.
Author Response
COVER LETTER
09 May 2022
Editor-in-Chief
Journal of Personalized Medicine
Dear Editor-in-Chief
We would like to express our sincere thanks to you and the reviewers for the thorough review of our manuscript (Manuscript ID: jpm-1688832) titled “Clinical utility of personalized serum IgG subclass ratios for the differentiation of IgG4-related sclerosing cholangitis (IgG4-SC) from primary sclerosing cholangitis (PSC) and cholangiocarcinoma (CCA)” and for the opportunity to submit a revised and improved version. We believe that by addressing the concerns, we have considerably improved our manuscript. Below this letter, we have provided point-by-point responses to the reviewer’s comment.
We hope that you find the current version of the manuscript suitable for publication in your journal. We will certainly be willing to make additional changes should they be required.
Thank you for your consideration. I look forward to hearing from you.
Sincerely,
Jong Kyun Lee, MD, PhD
Departments of Medicine, Samsung Medical Center, Sungkyunkwan University School
of Medicine, 81 Irwon-ro, Gangnam-gu, Seoul, 06351, South Korea
Email: jongk.lee@samsung.com
+82-2-3410-3409
82-2-3410-6983
Response to Reviewer Comments
Reviewer 2
The manuscript written by Jae Keun Park et al. investigates the best combinations of serum IgG subclass and IgG4 level for differentiating IgG4-SC from PSC or CCA. There are some interesting results where authors claimed that serum IgG4/IgG1 or IgG4/(IgG1+IgG3) level may help to differentiate IgG4-SC from PSC, and IgG4 alone is the most accurate serologic marker for the differentiation of IgG4-SC from CCA.
Regardless of the fact that it is an interesting work, some changes should be made in manuscript.
Comment 1.
In Statistical analysis the authors claimed that they used Kruskal-Willis test. This is not correct, it is Kruskal-Wallis test.
Response: Thank you for your feedback. We apologize for the confusion. We revised the Materials and Methods section of our manuscript as follows;
Revised Text:
2.3. Statistical analyses
We used the Kruskal-Wallis test to compare categorical and continuous data between groups. Receiver operator characteristic (ROC) curves were used to confirm optimal cutoff values of IgG subclass combinations and serum IgG4 to differentiate IgG4-SC from PSC or CCA. (Page 3, Line 100-103)
Comment 2.
Also, the main limitation of this study is the small number of the study patients. For serious conclusion it is necessary to include a significantly larger number of patients.
Response: Thank you for raising this point. Our thoughts are in line with you. As we mentioned in the limitations of our study, one of the limitations was the small number of the study patients, especially patients with PCS (n=27). The reason for the small number of patients with PCS was because PSC is rare in East Asia. We also planned the further large-scale multicenter-based prospective studies to confirm the suitability of the use of these combinations of serum IgG subclass and IgG4 level and our results. We revised the limitations and future works in the Discussion section of our manuscript as follows;
Revised Text:
“4. Discussion” section
The second limitation was the small number of the study patients, especially patients with PCS (n=27). The reason for the small number of patients with PCS was that PSC is rare in East Asia [29,30]. Therefore, to overcome the above limitation, a nationwide multi-center study is necessary. (Page 10, Line 325-328)
29 Li, P.; Chen, H.; Deng, C.; Wu, Z.; Lin, W.; Zeng, X.; Zhang, W.; Zhang, F.; Li, Y. Establishment of a serum IgG4 cut-off value for the differential diagnosis of IgG4-related disease in Chinese population. Modern Rheumatology 2016, 26, 583-587.
- Ang, T.L.; Fock, K.M.; Ng, T.M.; Teo, E.K.; Chua, T.S.; Tan, J.Y.L. Clinical profile of primary sclerosing cholangitis in Singapore. Journal of gastroenterology and hepatology 2002, 17, 908-913.
Further large-scale multicenter-based prospective studies are also required to confirm the suitability of the use of these combinations of serum IgG subclass and IgG4 level and our results. (Page 10, Line 339-341)
Comment 3.
It is necessary to correct the grammar of the English language.
Response: Thank you for your feedback. This manuscript has undergone English language editing by MDPI. The text has been checked for correct use of grammar and common technical terms, and edited to a level suitable for reporting research in a scholarly journal. MDPI uses experienced, native English-speaking editors. Full details of the editing service can be found at â–º https://www.mdpi.com/authors/english.
Revised Text:
Submitted Manuscript english-edited-jpm-1688832
Comment 4.
The results are very difficult to follow. Also, it is necessary to compare some additional clinical and biochemical data of patients for some serious conclusion.
Response:
- Thank you for this comment. Our thoughts are in line with you. We revised the result section as follows for the better following the results of study;
Revised Text:
- Results
3.1.1. Clinical characteristics of the study population
In Ttable 1, the clinical characteristics of the study population are listed. A total of 31 IgG4-SC, 27 PSC, and 40 CCA patients who checked an IgG subclass at diagnosis were included in this study. The Mmean age of diagnosis was significantly different among the 3 three groups. The mean age was 65.0 ± 11.9 years in the IgG4-SC group, 50.3 ± 13.4 years in the PSC group, and 63.5 ± 8.8 years in the CCA group (p = 0.008; Kruskal-Wallis test). Sex did not show a statistically significant difference. Except for IgG3, all IgG subclass showed statistically significant differences among the 3 three groups. The Mmean IgG1 (mg/dL) level was 962.1 ± 460.4 in the IgG4-SC group, 846.7 ± 327.0 in the PSC group, 654.9 ± 239.8 in the CCA group (p = 0.001; Kruskal-Wallis test). The Mmean IgG2 (mg/dLl) level was 743.8 ± 343.9 in the IgG4-SC group, 586.2 ± 252.3 in the PSC group, 511.0 ± 198.2 in the CCA group (p = 0.007; Kruskal-Wallis test). The Mmean IgG3 (mg/dl) level was 80.5 ± 79.8 in the IgG4-SC group, 74.2 ± 47.8 in the PSC group, 42.5 ± 25.8 in the CCA group (p = 0.053; Kruskal-Wallis test). The Mmean IgG4 (mg/dl) level was 226.5 ± 184.2 in the IgG4-SC group, 37.3 ± 20.1 in the PSC group, 46.5 ± 48.7 in the CCA group (p < 0.001; Kruskal-Wallis test). Elevated IgG4 levels (>135 mg/dL) levels were noted in 17 patients (54.8%) in the IgG4-SC group, and 2 two patients (5%) in the CCA group. The PSC group patients did not show elevated IgG4 levels (p < 0.001; Kruskal-Wallis test).
3.1.2. Performance of the serum IgG subclass combinations for the differential diagnosis of IgG4-SC from PSC
The comparison of the diagnostic performance of each IgG subclass combination between IgG4-SC and PSC was is demonstrated in Tttable 2. The A serum IgG4 level of ≥ 135mg/dL yieldeds a sensitivity of 54% (95% CI: 37-70) with a specificity of 100% (95% CI: 74-100) for IgG4-SC. The PPV was 100% (95% CI: 81-100), whereas the NPV was 44% (95% CI: 26-62). In this study, we analyzed the optimal cutoff values of serum IgG4 (cutoff values of serum IgG4 was 68 mg/dL). The An IgG4 ≥ 68mg/dL yieldeds a sensitivity of 64% (95% CI: 46-78) with a specificity of 100% (95% CI: 74-100) for IgG4-SC. PPV was 100% (95% CI: 83-100), whereas NPV was 50% (95% CI: 30-69).
We compared IgG4/subclass ratios to distinguish IgG4-SC from PSC by using ROC curves. The IgG4/subclass ratios consisted of the following groups: IgG4/IgG1, IgG4/(IgG1+IgG3), and (IgG4+IgG2)/(IgG1+IgG3). The largest areas under the curve (AUC) and reached the optimal combination of sensitivity and specificity at 0.087, 0.081 and 1.159 in IgG4/IgG1, IgG4/(IgG1+IgG3), and (IgG4+IgG2)/(IgG1+IgG3), respectively. In terms of accuracy, the IgG4/IgG1 and IgG4/(IgG1+IgG3) groups were found to be the highest group with the highest in terms of sensitivity, 70% and specificity 100%, and with statistically significantly higher sensitivity and accuracy than IgG4 ≥ 135 mg/dL (p=0.025).
3.1.3. Performance of the serum IgG subclass combinations for the differential diagnosis of IgG4-SC from CCA
The comparison of the diagnostic performance of each IgG subclass combination between IgG4-SC and CCA was is demonstrated as Ttable 3. The A serum IgG4 level of ≥ 135 mg/dL yieldeds a sensitivity of 54% (95% CI: 37-70) with a specificity of 95% (95% CI: 83-98) for IgG4-SC. The PPV was 89% (95% CI: 68-97), whereas the NPV was 73% (95% CI: 59-83). We analyzed the ROC curve to confirm the optimal cutoff values of serum IgG4 (cutoff values of serum IgG4 was 52 mg/dL). The IgG4 ≥ 52 mg/dL yieldeds a sensitivity of 80% (95% CI confidence interval: 63-90) with a specificity of 82% (95% CI: 68-91) for IgG4-SC. The PPV was 78% (95% CI: 61-88), whereas the NPV was 84% (95% CI: 70-92). Comparing the IgG4/subclass ratios to distinguish IgG4-SC from CCA by using the ROC curves, the largest AUC and reached the optimal combination of sensitivity and specificity at 0.087, 0.081 and 1.159 in IgG4/IgG1, IgG4/(IgG1+IgG3) and (IgG4+IgG2)/(IgG1+IgG3), respectively. In terms of accuracy, IgG4 ≥ 52 mg/dL was found to be the highest group with the highest sensitivity, 80%, and a statistically significantly higher sensitivity than IgG4 ≥ 135 mg/dL (p = 0.005). However, the accuracy of IgG4 ≥ 52mg/dL was not statistically significant compared with to IgG4 ≥ 135mg/dL (p = 0.405). Plus, the specificity of the IgG4 ≥ 52mg/dL was much lower than that of IgG4 ≥ 135mg/dL (IgG4 ≥ 52mg/dL vs. IgG4 ≥ 135mg/dL; 82% (95% CI: 68-91) vs. 95% (95% CI: 83-98), p = 0.025).
3.1.4. ROC curves of the IgG subclass combinations for the diagnosis of IgG4-SC from PSC
Table 4 and Figure 1A show the AUCs (95% CIs) of the serum IgG subclass combination for differentiations of IgG4-SC from PSC. IgG4 ≥ 68mg/dL, IgG4/IgG1 ≥ 0.087 and IgG4/(IgG1+IgG3) ≥ 0.081 were almost equally predictive of IgG4-SC. (IgG4+IgG2)/(Ig G1+IgG3) ≥ 1.159 had the lowest AUC for IgG4-SC (AUC: 0.730;, 95% CI:0.572-0.888), while IgG4/(IgG1+IgG3) ≥ 0.081 showed the highest AUC value for IgG4-SC (AUC: 0.853;, 95% CI:0.739-0.967). The AUCs of IgG4 ≥ 68 mg/dL and IgG4/IgG1 ≥ 0.087 were 0.827 (95% CI: 0.704-0.950) and 0.848 (95% CI: 0.732-0.963), respectively.
3.1.5. ROC curves of the IgG subclass combinations for the diagnosis of IgG4-SC from CCA
The AUCs (95% CIs) of the serum IgG subclass combinations for the differentiation of IgG4-SC from PSC are presented in Ttable 5 and Figure 1B. In our study, IgG4 ≥ 52mg/dL had the highest AUC values for IgG4-SC (AUC: 0.826;, 95% CI: 0.717-0.935), while (IgG4+IgG2)/(Ig G1+IgG3) ≥ 1.159 had the lowest AUCs for IgG4-SC (AUC: 0.617;, 95% CI: 0.478-0.757). IgG4/IgG1 ≥ 0.087 and IgG4/(IgG1+IgG 3) ≥ 0.081 were almost equally predictive of IgG4-SC. The AUCs of IgG4/IgG1 ≥ 0.087 and IgG4/(IgG1+IgG 3) ≥ 0.081 were 0.760 (95% CI: 0.639-0.881) and 0.753 (95% CI: 0.629-0.876), respectively.
- In our study, we aimed to find the best combinations of serum IgG subclass and IgG4 level for differentiating IgG4-SC from PSC or CCA. And this study is the retrospective designed single center study. Several additional data was gained retrospectively by reviewing the medical records. However, due to some missing values in the clinical and biochemical data collected retrospectively at the beginning of the study, direct comparison between diseases did not have statistical significance. Despite the anticipated weakness, this study offers some insight into finding the best combination of serum IgG subclass and IgG4 level for differentiate the IgG4-SC from PSC or CCA. We also planned the further large scale multicenter based prospective studies to overcome these limitations. We added and revised the limitation in the Discussion section as follows;
Revised Text:
“4. Discussion” section
Fourth, this study focuses on the combinations of serum IgG subclasses and IgG4 levels due to missing values in the clinical and biochemical data, but this is inherent to the retrospective study. Well-designed randomized prospective studies are needed to overcome this limitation of this study. (Page 10, Line 331-335)
We thank the editor of Journal of Personalized Medicine and the reviewers once again for their constructive feedback. In addition to the revisions provided above, please note that various minor edits were made to correct grammatical errors in our manuscript.

Round 2
Reviewer 2 Report
The manuscript written by Jae Keun Park et al. investigates the best combinations of serum IgG subclass and IgG4 level for differentiating IgG4-SC from PSC or CCA. There are some interesting results where authors claimed that serum IgG4/IgG1 or IgG4/(IgG1+IgG3) level may help to differentiate IgG4-SC from PSC, and IgG4 alone is the most accurate serologic marker for the differentiation of IgG4-SC from CCA.
The changes we have sought have been made to the text in manuscript.